# LazyAttention: Efficient Retrieval-Augmented Generation with Deferred Positional Encoding

**Haocheng Xia** [1]   **Mihir Pamnani** [2]   **Hanxi Fang** [3]   **Supawit Chockchowwat** [4]   **Yongjoo Park** [1]

## Abstract

Key-value (KV) caching accelerates inference of large language models (LLMs) by reusing past computations for generated tokens. Its importance becomes even greater in long-context applications such as retrieval-augmented generation (RAG) and in-context learning (ICL). However, conventional KV caching embeds positional information directly into the cache, limiting its reusability. Existing solutions either restrict reuse to prefixes or require expensive memory materialization for positional re-encoding. We introduce LazyAttention, a novel attention mechanism that kernelizes deferred positional encoding to enable zero-copy, position-agnostic KV reuse. By adjusting positional encoding within attention kernels *on-the-fly*, LazyAttention resolves the materialization bottleneck, allowing a single physical KV copy to serve multiple logical requests at arbitrary positions. Leveraging attention kernels tailored for prefilling and decoding, our system achieves significant efficiency improvements: under skewed document distributions, it reduces time-to-first-token (TTFT) by $1.37\times$ and increases inference throughput by $1.40\times$ compared to the state-of-the-art Block-Attention, while maintaining comparable output quality.

## 1. Introduction

Retrieval-augmented generation (RAG) greatly improves the quality and timeliness of responses by enriching user queries with external data (Lewis et al., 2020; Guu et al., 2020; Asai et al., 2023; Ram et al., 2023; Gao et al., 2024; Kalra et al., 2024; Lan, 2024; Xiong et al., 2024). However, processing this external data remains a major bottleneck for achieving low-latency RAG, since answer generation can only begin once the data has been fully processed. This step scales poorly—the computational complexity grows quadratically with input length (Jiang et al., 2024; Hu et al., 2025; Tang et al., 2024; Acharya et al., 2025; Zhang et al., 2025). The overhead will further worsen as modern models support increasingly longer context windows (Gemini Team et al., 2024). While each retrieved document must be processed at least once, its results could be stored in a form that enables more flexible reuse. This motivates a careful investigation into *reusable components*.

Existing inference approaches use conventional key-value (KV) cache (Qin et al., 2025) as reusable components, but remain ineffective due to their *position-awareness*. That is, cached values are reusable only if the associated data (e.g., a document) appears in the same position as before. This restriction is considered by earlier caching techniques, which accordingly focus on the identification of exactly matching document sequences (Kwon et al., 2023a; Gim et al., 2024; Jin et al., 2024). However, the chance of observing an exact match is lower than that of encountering individual documents. Recent works (Lu et al., 2025; Ma et al., 2025) show that KV cache can be reused even for the documents appearing in different positions if their positional information is re-encoded. This enhances reusability but is still memory-inefficient because position re-encoding requires duplicating the KV cache. In-place updates can cause race conditions and incorrect outcomes if two prompts in the same batch share the same data. This limitation—position-awareness—makes caching ineffective. To illustrate, suppose document popularity follows a Zipf distribution and each document may appear in any of $D$ prompt positions. With a cache budget of $C$ physical KV entries, a position-agnostic cache can store the top $C$ documents, whereas a position-aware cache spends entries on positional variants and covers only about the top $\lfloor C/D \rfloor$ documents. The resulting hit-ratio gap is $\frac{\sum_{i=1}^{C} i^{-\alpha}}{\sum_{i=1}^{\lfloor C/D \rfloor} i^{-\alpha}}$, which is $2.86\times$ for $D = 20$ and $C = 100$ under a moderately skewed Zipf distribution; Section 2 gives the full analysis.

In this paper, we argue that reusable components should be *position-agnostic* to enable effective caching under limited memory. On modern GPUs, high-bandwidth mem-

---

[1]Siebel School of Computing and Data Science, University of Illinois Urbana-Champaign [2]Nexla [3]Amazon [4]Google. Correspondence to: Yongjoo Park <yongjoo@illinois.edu>.

*Proceedings of the 43rd International Conference on Machine Learning*, Seoul, South Korea. PMLR 306, 2026. Copyright 2026 by the author(s).

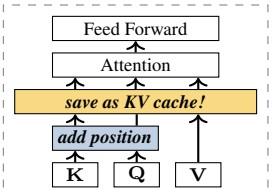
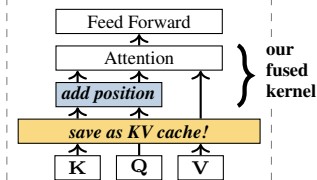

*(a)* Existing positional encoding. Positional encoding (blue box) is *eagerly* applied before saving (yellow box) the computed key (K) and queries (Q).

*(b)* Our positional encoding. Positional encoding (blue box) is *lazily* applied inside attention kernels and KV caching (yellow box) logically omits positions.

*Figure 1.* Comparison of positional encoding strategies within a Transform block. Existing methods, even the state-of-the-art Block-Attention for KV cache reuse, eagerly apply positions before caching, making KV caches position-dependent and non-shareable. LazyAttention applies positions lazily inside the fused attention kernel, enabling reuse and sharing without duplication.

ory (HBM) remains a scarce resource for LLM serving, as KV cache size grows with both context length and the number of concurrent requests. Existing memory-saving techniques—including KV compression (Ge et al., 2024; Liu et al., 2024a), grouped-query attention (Ainslie et al., 2023), and latent KV representations (DeepSeek-AI et al., 2024)—reduce the size of individual KV entries. However, they do not address the capacity wasted on storing position-specific variants of the same reusable content. A position-agnostic mechanism removes this redundancy: full coverage for $N$ reusable documents requires only $O(N)$ document-level KV entries rather than multiple position-specific copies per document. Under Zipf-distributed workloads, this lets the same memory budget cover more high-probability documents and translates into the distribution-dependent cache-hit improvement analyzed in Section 2. This would also make it feasible to preprocess and fully cache a medium-sized database using a combination of GPU HBM and host memory, enabling near-zero latency for a wide range of RAG applications. However, this possibility is nearly impossible with existing mechanisms. Positions are *eagerly* embedded into KV cache for nearly all models (e.g., BERT (Devlin et al., 2019), Llama (Dubey et al., 2024), DeepSeek (DeepSeek-AI et al., 2024)) running on the latest serving systems (e.g., vLLM (Kwon et al., 2023b), Orca (Yu et al., 2022), SGLang (Zheng et al., 2024)). Achieving position-agnostic reuses will therefore require a fundamentally different approach.

We propose LazyAttention, an attention mechanism that can enable *position-agnostic* KV reuse, significantly improving cache effectiveness for RAG applications. While the high-level concept of decoupling rotary position embedding (RoPE) from KV storage has been explored (Lu et al., 2025; Ma et al., 2025), prior attempts faced a critical *memory-compute trade-off*: they either required materializ-

ing position-adjusted copies (high memory/bandwidth cost) or restricted reuse to prefixes causing more recomputation (high computation cost). LazyAttention resolves this trade-off by *kernelizing* the RoPE-decoupling technique. The key idea shown in Figure 1 is to develop an alternative form of KV cache that can be shared among multiple prompts *without* approximating attention computation. LazyAttention achieves this by *deferring*, rather than omitting, positional encoding until the final stage—when the attention is computed. At that point, the kernel dynamically encodes positions on chip by considering relative distances between each query-key pair, requiring only a few additional kernel variables. Importantly, positional encoding in LazyAttention is only *transient*, lasting only for the brief period of attention computation. This design completely avoids additional data materialization in HBM. Despite the difference, LazyAttention's attention is mathematically identical to existing methods that duplicate KV caches for position re-encoding (Lu et al., 2025; Ma et al., 2025). As a result, it produces the same attention scores and generated answers.

LazyAttention is implemented based on vLLM (Kwon et al., 2023b) with two custom attention kernels for prefilling and decoding in Triton (Tillet et al., 2019). Designing separate kernels is non-trivial, as prefilling is typically compute-bound while decoding is memory-bandwidth-bound. To address these distinct bottlenecks, our kernels are carefully optimized to minimize extra computation in prefilling and extra I/O in decoding. Furthermore, our custom attention mechanism integrates positional encoding directly into the key-value matrix multiplications. Despite this added logic, the runtime overhead remains around $0.2\%$ in practice, while substantially improving cache effectiveness and reducing latency.

We evaluated LazyAttention against diverse baselines, including CacheBlend (Yao et al., 2025), Prompt Cache (Gim et al., 2024), and Block-Attention (Ma et al., 2025), on standard RAG benchmarks, demonstrating consistent improvements in latency and throughput empirically. Our approach is particularly effective in common RAG scenarios with skewed document access patterns, where hot documents are frequently reused: compared to the state-of-the-art Block Attention, LazyAttention reduces time-to-first-token (TTFT) by $1.37\times$ and increases inference throughput by $1.40\times$, while maintaining similar output quality. While our primary evaluation focuses on RAG, LazyAttention also benefits any workload where text chunks recur across requests, such as few-shot in-context learning (Appendix C.7) and parallel hypothesis agents (Fang et al., 2025). Our contribution can be summarized as follows.

- We propose a novel attention mechanism that kernelizes the RoPE-decoupling technique to enable zero-copy, position-agnostic KV reuse. By deferring positional encoding to a transient step within the attention kernel, our

method resolves the memory–compute trade-off that previously hindered the scaling of arbitrary-position reuse.

- We demonstrate that this position-agnostic design significantly increases the cache hit ratio. A single cached document entry can be shared across all requests, regardless of its position, maximizing cache efficiency under memory constraints. For skewed access patterns, the hit ratio improves by $7.5\times$ compared to prefix caching.

- We provide highly optimized Triton kernels [1] for both prefilling and decoding that implement our mechanism with negligible overhead (around 0.2%) even for long-context inputs, translating our architectural improvements into substantial practical gains in end-to-end throughput and latency for RAG workloads.

## 2. Background and Motivation

In this section, we briefly review the literature of RAG and KV cache reuse, then discuss why existing techniques struggle with efficiency under dynamic contexts.

Retrieval-augmented generation (RAG) (Lan, 2024; Guu et al., 2020; Lewis et al., 2020; Chan et al., 2025; Zhang & Park, 2025) augments LLMs with external knowledge. Given a query $\mathcal{Q}$, a retriever selects top-$N$ relevant documents $\mathcal{D} = \{d_1, \ldots, d_N\}$, which the LLM conditions on when producing an answer. The most common approach is simple concatenation of retrieved documents and the query (Izacard & Grave, 2021), though more complex schemes exist (Borgeaud et al., 2022).

RAG enables flexible knowledge grounding but also introduces variability: newly collected documents may need to be inserted while irrelevant or low-quality documents may need to be removed across queries (Li et al., 2023; Yoran et al., 2024; Cuconasu et al., 2024). Such changes alter the prefix, forcing standard attention to re-run the costly prefilling step. This motivates techniques for reusing previously computed KV caches beyond the prefix to support dynamic contexts efficiently.

Existing methods, including Prompt Cache (Gim et al., 2024), CacheBlend (Yao et al., 2025), EPIC (Hu et al., 2025), TurboRAG (Lu et al., 2025), and Block-Attention (Ma et al., 2025), all rely on re-encoding positions. While effective to some extent, this coupling between cache and position leads to inefficiency: each distinct document position can consume a separate KV-cache copy. In short, existing methods either suffer from low reusability or exhaust memory with redundant cache copies.

**Hit-ratio impact under limited cache capacity** We now quantify why position-awareness directly lowers cache hit ratio. Assume reusable documents follow a Zipf distribution, where the $i$-th most popular document has probability proportional to $i^{-\alpha}$. If each document may appear in any of $D$ prompt positions and the cache stores $C$ physical KV entries, a position-agnostic cache stores one entry per document and caches the top $C$ documents:

$$H_{\text{agnostic}}(C) = \frac{\sum_{i=1}^{C} i^{-\alpha}}{\sum_{j=1}^{N} j^{-\alpha}}, \tag{1}$$

where $N$ is the number of reusable documents. A position-aware cache must instead spend capacity on positional variants. If each cached document needs up to $D$ position-specific copies, the same budget covers only about $\lfloor C/D \rfloor$ unique documents:

$$H_{\text{aware}}(C, D) \approx \frac{\sum_{i=1}^{\lfloor C/D \rfloor} i^{-\alpha}}{\sum_{j=1}^{N} j^{-\alpha}}. \tag{2}$$

The resulting hit-ratio advantage is

$$\frac{H_{\text{agnostic}}(C)}{H_{\text{aware}}(C, D)} \approx \frac{\sum_{i=1}^{C} i^{-\alpha}}{\sum_{i=1}^{\lfloor C/D \rfloor} i^{-\alpha}}. \tag{3}$$

For example, when documents may appear in $D = 20$ positions and the cache can store $C = 100$ KV entries, Equation (3) gives a $2.86\times$ hit-ratio advantage under a moderately skewed Zipf distribution. The gap arises under the same physical memory budget because position-aware caching spends entries on duplicate positional variants, while position-agnostic caching covers more unique documents. This raises the central question: *can we achieve high cache hit ratios under limited memory without materializing position-specific KV copies?*

Relative positional encodings such as RoPE (Su et al., 2024) suggest a possible remedy: keys can be rotated on-the-fly to realign with the new positions of reused documents. However, applying rotations dynamically during decoding requires recomputing large key matrices for every token, inducing prohibitive overhead that violates the strict latency constraints of real-time inference.

## 3. LazyAttention: Algorithm and Analysis

In this section, we show how the positional information of reused documents can be adjusted *during* the attention calculation. Then we analyze the cost of deferred positional encoding for prefilling and decoding, respectively. Inspired by the analysis, we present how to integrate LazyAttention with FlashAttention (Dao et al., 2022; Dao, 2024; Shah et al., 2024) seamlessly to achieve efficient computation.

**Problem statement** Motivated by the limitations of existing approaches and the constraints of GPU memory, the

---

[1] https://github.com/illinoisdata/lazy-attention

problem is to develop a method that efficiently manages and manipulates the positional information of documents in KV caches *on the fly* during attention computation, without incurring high recomputation or extensive copying costs. The method should correctly handle the global positional alignment required by the request and support scenarios involving multiple requests accessing the same document at different global offsets within a batch.

### 3.1. Deferred Positional Encoding

In the standard Transformer architectures (Vaswani et al., 2017), positional encoding is applied before attention computation, forcing explicit duplication of the KV cache whenever document positions differ. Our approach logically *defers* positional encoding until the attention computation step, ensuring the KV cache remains position-agnostic.

**Standard attention**  In the standard attention formulation, given query, key, and value matrices, $\mathbf{Q}$, $\mathbf{K}$, and $\mathbf{V}$, the attention output is computed as follows,

$$\text{Attention}(\mathbf{Q}, \mathbf{K}, \mathbf{V}) = \text{softmax}\left(\frac{\mathbf{Q}\mathbf{K}^\top}{\sqrt{d_k}}\right)\mathbf{V}, \quad (4)$$

where $d_k$ is the dimensionality of $\mathbf{K}$.

Positional encoding allows the model to exploit sequence order when computing attention scores, e.g., allocate more attention to the recent context. A variety of methods exist—ranging from absolute to relative encodings such as XLNet-style (Yang et al., 2019), rotary positional embedding (RoPE) (Su et al., 2024), and interleaved rotary position embedding (Llama Team, 2025). Among these, RoPE has become the most widely adopted due to its simplicity and effectiveness. In this paper, we focus on the original RoPE and its variants (Llama Team, 2024; Liu et al., 2024b). With RoPE, Equation 4 becomes:

$$\text{Attention}(\mathbf{Q}, \mathbf{K}, \mathbf{V}) = \text{softmax}\left(\frac{f(\mathbf{Q}, \mathbf{m})f(\mathbf{K}, \mathbf{n})^\top}{\sqrt{d_k}}\right)\mathbf{V},$$

where $f$ denotes the positional encoding function, and $\mathbf{m}$, $\mathbf{n}$ represent the positional indices of the tokens in $\mathbf{Q}$ and $\mathbf{K}$.

**Deferred encoding**  In our method, the query, key, and value matrices are first computed from the input sequence without positional information. Positional encoding is applied only *during* attention computation, after retrieving entries from the KV cache. This means the cache is regarded as purely content-based keys and values, which can be reused across arbitrary positions. The positional adjustments are applied at runtime, yielding both mathematical correctness and significantly improved cache efficiency, as shown in the following example.

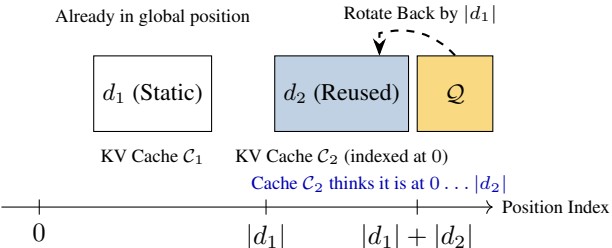

*Figure 2.* Illustration of Example 3.1.

**Example 3.1.** *Suppose a query $\mathcal{Q}$ encodes its positional index relative to the document length. Consider two documents, $d_1$ and $d_2$, with independently generated KV caches $\mathcal{C}_1$ and $\mathcal{C}_2$, both indexed from position $0$. To reuse $\mathcal{C}_2$ for $\mathcal{Q}$ immediately following the processing of $d_1$, we rotate $\mathcal{Q}$ backward by $|d_1|$ before the attention computation. This adjustment ensures positional consistency by aligning the query's phase with the pre-computed cache.*

**Naive deferred encoding is expensive**  Following the idea of existing reuse approaches (Ma et al., 2025; Lu et al., 2025), one straightforward implementation for deferred encoding is shown in Figure 3(a). Specifically, whenever we load a KV block, we first reset its position by rotating the beginning of $\mathbf{K}$ back to position $0$ and then rotate it forward to the target position.[2] Now we analyze its cost using tiling. Focus on the computation for a set of Q, K, and V tiles, where the shape of a query tile is $(M, D)$ and the shape of a K/V tile is $(N, D)$. We note that $M$ is the number of tokens in a Q tile, $N$ is the number of tokens in a K/V tile, and $D$ is the head size. Then the main cost is from two general matrix multiplications (GEMMs), i.e., $\mathbf{Q}\mathbf{K}^\top$ and $\mathbf{P}\mathbf{V}$ where $\mathbf{P} = \text{Softmax}(\frac{\mathbf{Q}\mathbf{K}^\top}{\sqrt{d_k}})$ and $d_k$ is the dimension of key ($d_k = D$ here). We always perform these two GEMMs with $4MND$ floating point operations (FLOPs).[3] Applying RoPE requires a 2D rotation for every pair of feature dimensions, incurring 6 FLOPs per pair (or 3 FLOPs per scalar), derived from the 4 multiplications and 2 additions in the rotation matrix application. Therefore, *two* rotations on a K tile would introduce $\Delta\text{FLOPs}_{\text{naive}} = 2 \times 3ND = 6DN$ FLOPs. We have the relative extra cost is,

$$\frac{\Delta\text{FLOPs}_{\text{naive}}}{\text{FLOPs}_{\text{baseline}}} \approx \frac{6DN}{4MND} = \frac{3}{2M}.$$

For *decoding*, where $M = 1$ as LLMs generate token one by one, then the deferred rotation introduces 150% extra FLOPs. For *prefilling* with $M = 128$ (default in vLLM when prefilling), the extra FLOPs are still 1.17%. On the I/O side, each rotation needs a $\cos/\sin$ vector of length $D$

---

[2]Equivalently, two successive rotary transforms per block.

[3]We count multiply and add as two FLOPs.

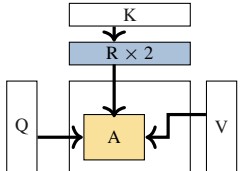

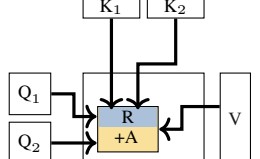

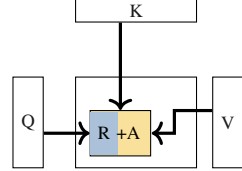

*(a)* Existing: keys are rotated (R) twice before attention (A)

*(b)* Our prefilling: keys are rotated only once inside the fused attention kernel

*(c)* Our decoding: query is rotated inside the fused attention kernel

*Figure 3.* Comparison of Rotary Embedding (R) placement in tiled attention. *(a)* Naive deferred encoding: Rotates keys twice (resetting to zero, then shifting to target) before attention (A), incurring high computational overhead. *(b)* LazyAttention (prefilling): Rotates only the keys inside the attention kernel on the fly, allowing Keys/Values to be read as-is to avoid materialization overhead ($K_1/K_2$ denote the RoPE half-dimensions). The rotary vector is loaded from HBM to avoid extra computation. *(c)* LazyAttention (decoding): Fuses rotation directly into the attention inner loop, applying it to Q on-the-fly via relative offsets, ensuring the KV cache remains position-agnostic with negligible cost. The rotary vector is computed on the fly to save bandwidth.

per token, hence two rotations add $2ND$ elements. Relative to reading three tiles, the extra bandwidth fraction is

$$\frac{\Delta \text{I/O}_{\text{naive}}}{\text{I/O}_{\text{baseline}}} \approx \frac{2ND}{MD + 2ND} = \frac{2N}{M + 2N}.$$

For *decoding* with $N = 16$ (default in vLLM when decoding), the extra IO would be approaching 100%, which is unaffordable. We note that the overhead of loading a Q tile is usually amortized since it can be used for multiple K tiles.

**Why this gets amplified on GPUs**   Modern attention kernels are often optimized to operate close to hardware limits, making them highly sensitive to modifications in the inner loop. Additional work, especially uncoalesced memory accesses, can introduce non-negligible overhead. Therefore, even seemingly minor changes, such as adding scalar loads to the decoding critical path, may noticeably degrade latency due to pipeline stalls and reduced warp occupancy.

Given the high overhead incurred by a naive implementation, we must find an efficient way to implement deferred positional encoding with low overhead.

### 3.2. Efficient Rotation with Tiling

We implement LazyAttention within attention kernels via careful system-algorithm co-design to reduce the overhead. First, we review a key property of RoPE.

**Fact 3.2** (Relative rotation for RoPE). *RoPE can be represented as a rotation matrix $\mathbf{R}$ applied to queries and keys, with the rotation angle determined by token positions. For any token pair from $\mathbf{Q}$ and $\mathbf{K}$ with positions $\mathbf{m}$ and $\mathbf{n}$,*

$$(\mathbf{R_m q})^\top (\mathbf{R_n k}) = \mathbf{q}^\top \mathbf{R_m^\top R_n}\, \mathbf{k} = \mathbf{q}^\top \mathbf{R_{n-m}}\, \mathbf{k},$$

*i.e., attention depends only on the* relative *offset* $\mathbf{n} - \mathbf{m}$.

**Rotate Q or K**   Since only relative offset matters, to change the relative position between $\mathbf{Q}$ and $\mathbf{K}$, we can rotate either $\mathbf{Q}$ or $\mathbf{K}$. We note that rotate both of them is

unnecessary and increases overhead. The choice of which nucleus to spin depends largely on the pattern of attention computation kernels.

**Prefilling (compute-bound)**   Due to the design of PagedAttention (Kwon et al., 2023b), prefilling kernels typically have a relative long Q tile and a short K tile, e.g., $M = 128$ and $N = 16$, hence *rotating* $\mathbf{K}$ is cheaper than rotating $\mathbf{Q}$. We apply a *single* relative rotation with offset $\Delta$ (never "back-to-zero then forward") like Figure 3(b), combining the two half dimensions as

$$k_1' = k_1 \cos \Delta - k_2 \sin \Delta, \qquad k_2' = k_1 \sin \Delta + k_2 \cos \Delta.$$

For computation, this introduces 3 FLOPs per scalar of a K tile. Compared with the baseline $\text{FLOPs}_{\text{baseline}} = 4MND$, the relative overhead factor of prefilling with deferred encoding is

$$\epsilon_{\text{prefilling}} = \frac{3ND}{4MND} = \frac{3}{4M}. \tag{5}$$

When $M = 128$, $\epsilon_{\text{prefilling}} = 0.59\%$. To save bandwidth, we keep a local position for each document's KV cache, i.e., indexing from position $0$. Therefore, when reusing a document, a fixed offset can be used for the whole document. Then we table-drive $\cos / \sin$ and load a $D$-length row once per document, adding small additional bandwidth $\approx \frac{D}{2DN} = \frac{1}{2N}$. As shown in Figure 3(b), we further divide each tile into two pieces for convenient rotation. Here, we omit the overhead of loading a Q tile. The overhead ratio for bandwidth will not exceed 3.1%. We note that since the iteration of K tiles is in the inner loop (Dao, 2024), rotating $\mathbf{K}$ would not introduce a high register pressure, which can be crucial for compute-bound kernels.

**Decoding (bandwidth-bound)**   For decoding, the attention computation kernel has a different mode because the length of a q tile is strictly $1$. While rotating a K tile would cost $3ND$ FLOPs, rotating a Q tile would cost a fixed $3D$ FLOPs. Besides, we *rotate* $\mathbf{Q}$ only when encountering different documents as shown in Figure 3(c). Let $r$ be the

fraction of KV tiles whose relative offset changes, and $B$ be the number of blocks in a document, $r = \frac{1}{B}$. Rotating a singe Q tile costs $3MD$, giving per-trigger overhead $\frac{3MD}{4MND} = \frac{3}{4N}$. When $N = 64$, the relative overhead factor is 1.17%. Averaged across all tiles, this becomes

$$\epsilon_{\text{decode}} = r \cdot \frac{3}{4N} = \frac{3}{4BN}. \tag{6}$$

This overhead becomes negligible for large $B$; for example, with documents exceeding 1,600 tokens, $r \leq 1\%$ and $\epsilon_{\text{decode}} \leq 0.01\%$. To eliminate inner-loop memory traffic, we bit-pack (`block_id`, `offset`, `mask`) into a single 64-bit register, unpacking metadata via register shifts to bypass global loads. For severe I/O intensive scenarios, we can further eliminate memory accesses by computing $\cos/\sin$ rotary vectors on-the-fly. Finally, we avoid partitioning the Q tile like the prefilling kernel shown in Figure 3(b), as splitting such a small tile would significantly degrade GEMM efficiency.

### 3.3. Analysis: Complexity of LAZYATTENTION

**Compute** Regarding computational complexity, LazyAttention preserves the asymptotic bounds of standard attention. Given sequence length $L$ and head dimension $D$, a single head performs $\mathcal{O}(L^2D)$ FLOPs for the two core GEMMs ($\mathbf{QK}^\top$ and $\mathbf{PV}$). The linear projections, which map the hidden dimension $d_{\text{model}}$ to the head dimension $D$, contribute $\mathcal{O}(Ld_{\text{model}}D)$ but remain unaffected by our design. Consequently, deferred rotation introduces only the constant-factor overheads derived earlier: $\frac{3}{4M}$ for prefilling and $\frac{3}{4BN}$ for decoding.

**Memory** Regarding memory and bandwidth, we maintain a *position-agnostic* KV cache logically, storing keys and values uniquely per token with $\mathcal{O}(LD)$ cost per head. In contrast, approaches that embed absolute positions must re-materialize cache entries for each shift, inflating memory usage proportional to the number of reused offsets. Decoupling position from KV data eliminates this redundancy, preserving a footprint linear in $L$.

### 3.4. Analysis: TTFT and Decoding Overhead

We analyze the overhead of LazyAttention on the two critical metrics of inference: Time-to-First-Token (TTFT) and Inter-Token Latency. Let $L$ be the total sequence length (including reused documents), $L'$ be the length of the new prompt segments, $D$ be the head size, $M$ be the query tile size, and $N$ be the KV tile size. We adopt a roofline model where a kernel's time cost is $T \approx \max(F/\mathcal{P}, B/\mathcal{B})$.

**TTFT** In modern serving systems (e.g., vLLM), TTFT is determined by the *prefilling kernel*, which processes

the prompt and generates the first token in a single pass. For RAG workloads, we only compute attention between the new prompt ($L'$) and the full context ($L$), resulting in $4LL'D$ FLOPs. Unlike the naive deferred approach which costs $6DN$ FLOPs (rotating $\mathbf{K}$ twice), LazyAttention performs a *single* relative rotation on $\mathbf{K}$ (since $N < M$ in prefilling). This incurs only $3ND$ extra FLOPs per tile pair. The relative computational overhead is $\epsilon_{\text{prefilling}} = \frac{3}{4M}$. Regarding I/O, loading one rotary vector adds negligible overhead ($\leq \frac{1}{2N}$ relative to reading $\mathbf{K}$ and $\mathbf{V}$). Thus, the TTFT is modeled as follows,

$$\text{TTFT} \approx T_{\text{prefilling}} \approx \max\left( \frac{4LL'D(1 + \frac{3}{4M})}{\mathcal{P}}, \frac{LD(2 + \frac{1}{N})}{\mathcal{B}} \right) \tag{7}$$

Note that the bandwidth term is dominated by reading the cached history from HBM ($\approx 2LD$), while the compute term scales with $L \cdot L'$. Equation 7 demonstrates that LazyAttention not only minimizes overhead ($\approx 0.59\%$ for $M = 128$) but also preserves the efficiency gains from document reuse (where $L' \ll L$).

**Decoding latency** For subsequent token generation, the system switches to the decoding kernel ($M = 1$), which is typically bandwidth-bound. LazyAttention fuses the rotation into the inner loop, rotating the query $\mathbf{q}$ only when the relative offset changes. Crucially, because rotation occurs in registers, the memory traffic remains at the baseline's $2LD$ bytes (reading only $\mathbf{K}$ and $\mathbf{V}$). Thus, the decoding latency remains at the optimal roofline as original attention computation,

$$T_{\text{decode}} \approx \frac{2LD}{\mathcal{B}}.$$

This confirms that LazyAttention introduces zero I/O overhead during generation.

### 3.5. Generalization Beyond Standard RoPE

LazyAttention is not tied to standard RoPE. It applies whenever positional effects can be injected into attention-score computation, rather than materialized as position-adjusted KV states. This covers the common relative-position cases used in modern LLM serving, while preserving exactness for identical cached chunks.

**RoPE-family variants** For RoPE-family variants, including interleaved RoPE used by Llama (Dubey et al., 2024), scaled/NTK variants, and YaRN (Liu et al., 2024b), the attention score can still be written as $\mathbf{q}^\top \mathbf{R}_{\mathbf{n}-\mathbf{m}} \mathbf{k}$ with variant-specific scaling metadata. The kernel therefore only needs token positions and lightweight encoding parameters; the overall zero-copy structure remains unchanged.

**GQA/MQA compatibility** Grouped-query attention (GQA) (Ainslie et al., 2023) and multi-query attention

(MQA) change the mapping from query heads to shared KV heads, but do not alter how each $\mathbf{q}$–$\mathbf{k}$ score is computed. Thus, LazyAttention requires no algorithmic change for GQA/MQA models.

**Score-space positional methods (ALiBi)** The same lazy principle extends to score-space relative-position methods such as ALiBi (Press et al., 2022), where the score is modified by a position-dependent bias rather than a rotation. Linear attention is outside our current scope because it changes the attention state representation rather than only the positional score computation. With a case study over Falcon-7B (Almazrouei et al., 2023), a model supports both RoPE and ALiBi, we validated that the extra decoding cost is less than 0.06%. This demonstrates the generalizability of LazyAttention accross different relative positional encoding methods.

# 4. Evaluation

In this section, we evaluate LazyAttention along four research questions (RQs) to demonstrate its advantages, analyze its overhead, and stress-test the scope highlighted by reviewers. More experimental details and additional generalization studies can be found in Section B.

**RQ1** Does LazyAttention reduce time for the first token (TTFT) under different serving loads?

**RQ2** For repeated documents across requests, does LazyAttention attain a high KV hit ratio with limited GPU memory for KV cache?

**RQ3** What is the latency overhead of the extra deferred rotation operation of LazyAttention in prefilling and decoding respectively?

**RQ4** Does LazyAttention preserve generation quality for different benchmark datasets?

**Implementation** We implement LazyAttention on the top of vLLM (vLLM Team, 2025) within 5K lines in Python based on PyTorch v2.7 and CUDA 12.4. Our implementation is designed to be compatible and non-intrusive with the existing framework, allowing for easy integration into various models. Specifically, we use the vLLM v0.8.5.post1 V1 for efficient inference and model management, using its capabilities to optimize memory usage and computational efficiency. The code is available at https://github.com/illinoisdata/lazy-attention.

**Models, Hardware, and Datasets** We evaluate LazyAttention using Tulu3-Block-FT[4] which is fine-tuned for

---

[4] https://huggingface.co/ldsjmdy/Tulu3-Block-FT

Block-Attention (Ma et al., 2025) from Llama-3.1-Tulu-3-8B-SFT on a machine with 120GB RAM, an NVIDIA H100 96GB GPU (GH200 chipset) (NVIDIA, 2022). To demonstrate generalization, we also extend our evaluation to Llama-3.1-70B-Instruct and Qwen3-8B, and test on NVIDIA A100 and A40 GPUs (details in Appendix C). Our evaluation uses four QA benchmarks: (1) *2WikiMQA* (Ho et al., 2020), which requires reading multiple paragraphs, each treated as a document in LazyAttention; (2) *HotpotQA* (Yang et al., 2018), a multi-hop dataset requiring reasoning across supporting documents; (3) *TriviaQA* (Joshi et al., 2017), a reading comprehension benchmark with long web-page contexts; and (4) *NarrativeQA* (Kočiský et al., 2018), where questions demand understanding long narratives such as novels and scripts.

**Baselines** We compare LazyAttention against the following baselines: (1) *Prompt Cache* (Gim et al., 2024), the standard RAG model using a fixed-length cached prefix; (2) *CacheBlend* (Yao et al., 2025), a masked variant of RAG that improves accuracy; (3) *Block-Attention (vLLM)* (Ma et al., 2025), a block-based mechanism for cache efficiency, re-implemented in vLLM for fairness; (4) *Prefix Caching* (Kwon et al., 2023a; vLLM Team, 2025), the standard prefix caching in vLLM; and (5) a *MEPIC-like* (Wang et al., 2025) reimplementation that stores fully NoPE KV and applies rotary offsets inside a fused attention operator (see Appendix C.9 for details).

## 4.1. Higher Responsiveness with Shorter TTFT

We evaluate system responsiveness by generating request streams with controlled arrival rates (req/s), sampling from a mixture of four datasets. Each request queries five documents whose KV states have been precomputed. We assess two distinct traffic regimes: *Uniform* sampling (representing low reuse potential) and *Skewed* sampling (Zipfian distribution with $\alpha = 2.1$, representing high reuse). All methods operate under identical batching policies, paging parameters, and memory budgets. We report mean TTFT and compare against key baselines, bounded by *Full Reuse* (theoretical lower bound) and *Full Recompute* (upper bound).

Under *Uniform* traffic (Figure 4a), where reuse opportunities are naturally scarce, LazyAttention remains highly competitive. It tracks *Block-Attn* closely at low-to-moderate loads and consistently outperforms *Prefix Caching*, *Prompt-Cache*, and *CacheBlend*. This performance validates our kernel analysis in Section 3.4: the prefilling overhead is minimal ($\frac{3}{4M}$) and the decoding overhead averages $r \cdot \frac{3}{4N}$. Since the fraction $r$ of nonzero-offset tiles is typically small and the tile size $N$ is adequate (e.g., 64), our method incurs negligible cost even when reuse is absent.

Under *Skewed* traffic (Figure 4b), where a small set of docu-

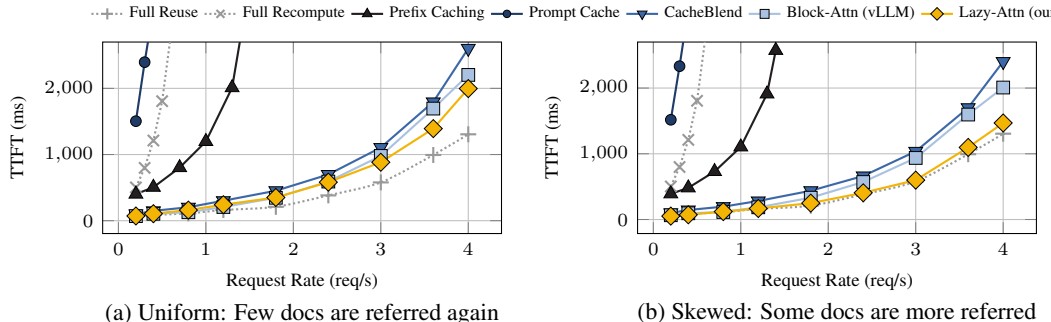

(a) Uniform: Few docs are referred again  (b) Skewed: Some docs are more referred

*Figure 4.* TTFT vs request rate under different document distributions. We vary the request rate (req/s) and plot time-to-first-token (TTFT).
(a) *Uniform*—few documents recur across requests (low reuse). (b) *Skewed*—a small set of documents is frequently reused (high reuse).

*Table 1.* VRAM cache hit ratio (%) under different KV cache budgets. A high hit ratio indicates more effective cache reuse. LazyAttention is the most effective, and its advantage persists up to the full GPU budget (a single-copy design keeps reusing documents at any offset, whereas the gains of position-coupled methods saturate).

| KV Cache Mem Size (→) | 1 GB | | | 5 GB | | | 10 GB | | | 50 GB | | | No-limit[†] | | |
|---|---|---|---|---|---|---|---|---|---|---|---|---|---|---|---|
| Document Skewness (→) | Low | Mid | High | Low | Mid | High | Low | Mid | High | Low | Mid | High | Low | Mid | High |
| Prefix Caching | 0.00 | 0.00 | 0.00 | 0.04 | 0.30 | 0.70 | 0.08 | 0.55 | 1.22 | 0.46 | 1.95 | 3.07 | 0.58 | 2.16 | 3.25 |
| CacheBlend | 1.51 | 4.95 | 5.96 | 6.29 | 14.43 | 14.95 | 8.91 | 17.33 | 16.78 | 14.32 | 21.87 | 19.06 | 15.21 | 22.45 | 19.36 |
| Block-Attention (vLLM) | 1.84 | 6.03 | 7.27 | 7.67 | 17.59 | 18.23 | 10.86 | 21.13 | 20.47 | 17.46 | 26.67 | 23.24 | 18.55 | 27.38 | 23.61 |
| *LazyAttention (Ours)* | **3.47** | **11.11** | **13.57** | **10.85** | **21.16** | **20.49** | **13.78** | **23.89** | **21.92** | **20.22** | **28.44** | **24.14** | **21.33** | **29.09** | **24.50** |

[†] No-limit: cache bounded only by GPU memory — NVIDIA H100 96 GB GPU (GH200 chipset) at 0.9 util, ≈66 GB usable KV for the 8B model. All cells are trace-driven simulation (Zipf doc popularity, skew $\alpha = 1.1/1.5/2.1$ for Low/Mid/High; 100k-doc pool, $K = 10$ docs/query).

ments is frequently accessed at varying positions, LazyAttention demonstrates superior scalability. We achieve significantly lower TTFT and sustain higher throughput before saturation. This advantage stems directly from *position-agnostic reuse*: LazyAttention utilizes the same physical KV blocks across different logical offsets without materializing new copies. In contrast, baselines face structural inefficiencies: *CacheBlend* incurs reconstruction overheads, *PromptCache* expands prompt length, and *Block-Attn* suffers from cache fragmentation due to storing duplicate, position-dependent KV copies. We observe consistent speedups on the larger Llama-3.1-70B model and across different hardware configurations (Appendix C.1).

## 4.2. Higher Cache Hit Ratios

We evaluate cache efficiency by varying both the KV cache budget (1/5/10/50 GB and a no-limit setting constrained only by available GPU memory) and document popularity skew (Low/Mid/High with $\alpha = 1.1/1.5/2.1$). Our primary metric is the VRAM cache hit ratio, defined as the fraction of KV-block lookups served directly from cache without recomputation. As shown in Table 1, LazyAttention achieves the highest hit ratio in every budget–skew configuration. The advantage is most pronounced—in relative terms—under tight memory, where avoiding position-coupled duplicates frees scarce VRAM for additional docu-

ments: at 1 GB, LazyAttention nearly doubles the strongest baseline (13.57% vs. 7.27% at high skew, 3.47% vs. 1.84% at low skew). As the budget grows, all methods improve and the gap narrows, yet LazyAttention retains the lead at every skew level—e.g., 23.89% versus 21.13% for Block-Attention at 10 GB and mid skew. Crucially, the benefit does not disappear once memory is abundant: even in the no-limit, full-GPU setting, LazyAttention leads at all skews (29.09% vs. 27.38% at mid skew, 21.33% vs. 18.55% at low skew). These results confirm that storing a single physical KV copy per document, reusable at arbitrary offsets, makes position-agnostic reuse valuable across the entire memory-budget spectrum—most decisively when VRAM is the binding constraint, and still beneficial when it is not.

## 4.3. Fused kernel efficiency: our rotation overhead is negligible

To isolate kernel-level costs, we construct a single long RAG request with five documents (each 4,096 tokens) and a 64-token query. The serving system uses a 2,048-token chunk budget per pass. We preload three documents' KV blocks in DRAM to emulate "hot" content and leave the remaining documents "cold." Common preamble is omitted. We compare two ablations: *w/o Deferred Rotation* (no position-agnostic reuse; blocks are processed in the conventional pipeline) and *with Deferred Rotation (ours)*, which reuses

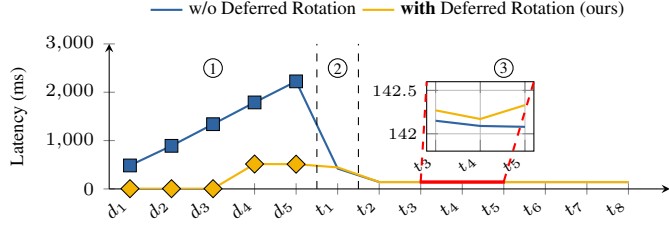
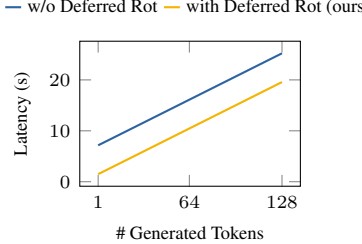

*(a)* Inference time over prefilling (d1–d5) and decoding steps (t1–t8)    *(b)* Cumulative generation time

*Figure 5.* Overhead analysis. (a) Breakdown of overhead across phases: ① document processing, ② query prefilling, and ③ decoding. Deferred rotation adds only 0.13% overhead in decoding. (b) Accumulated latency with and without deferred rotation, where the dominant gain comes from document processing rather than token generation.

*Table 2.* Question-answer accuracy for various benchmark datasets. *Exact match* scores are reported. Our LazyAttention performs the identical computations as Block-Attention, achieving nearly identical scores, where slight differences are due to tokenization and limited precision in floating-point operations. That is, LazyAttention significantly reduces TTFT and increases the reuse opportunities (as reported before) with negligible accuracy loss.

| Dataset | Full-Attn | CacheBlend | Block-Attn | *Block-Attn (vLLM)* | *Lazy-Attn (ours)* |
|---------|-----------|------------|------------|---------------------|---------------------|
| 2WikiMQA | 73.6 | 71.1 | 72.2 | 71.4 | 70.7 |
| TriviaQA | 75.2 | 69.2 | 72.3 | 72.1 | 73.0 |
| NarrativeQA | 62.2 | 60.1 | 60.4 | 61.0 | 59.7 |
| HotpotQA | 76.2 | 69.7 | 75.1 | 72.5 | 73.3 |
| Average | 71.8 | 67.5 | 71.2 | 69.3 | 69.2 |

cached KV at arbitrary offsets and performs the necessary RoPE rotation inside the fused attention kernel. Scheduler, batching, paging, and launch parameters are identical; only the rotation path differs.

As shown in Figure 5, we analyze the overhead across three key phases. ① Document processing: our method reduces latency to near-zero for hot documents by reusing KV cache without duplication, whereas recomputation costs dominate the baseline. ② Query prefilling: the performance remains comparable to the baseline; our kernel rotates keys just once per KV tile using the relative offset, introducing only a marginal $\frac{3}{4M}$ increase in FLOPs (Section 3.2). ③ Decoding: the latency curves overlap, with the zoomed inset revealing a negligible per-token overhead of *0.13%*. This aligns with our theoretical analysis of $r \cdot \frac{3}{4N}$ (Section 3.4), given the small fraction $r$ of nonzero-offset tiles. Figure 5 (b) confirms that the cumulative generation gap remains constant as the token count increases, proving that decoding overhead does not accumulate. We further verify that this efficiency holds for document lengths up to 16K and context lengths up to 128K (Appendix C.4).

### 4.4. Generation Quality

We evaluate LazyAttention following the setting of Block-Attention (Ma et al., 2025), assessing QA generation quality via Exact Match (EM). Unless otherwise specified, we use the same model checkpoint, prompts, and decoding hyper-

parameters. Empirically, LazyAttention attains EM scores comparable to Block-Attention, confirming that deferred positional encoding preserves model fidelity. Because we serve all methods on the same vLLM engine, the absolute EM scores sit slightly below those reported by Ma et al. (2025) under HuggingFace `transformers`; this offset arises from differences in numerical precision and attention-kernel implementation in vLLM rather than from any particular method, and it shifts all methods equally. We further validate LazyAttention on a long-form literature-review task and a few-shot classification workload, demonstrating that it accelerates recurring-chunk workloads beyond simple QA (Appendices C.7 and C.8).

## 5. Conclusion

In this paper, we proposed LazyAttention, a novel deferred positional encoding mechanism for Transformer models. Our approach decouples positional encoding from the KV cache, allowing for more efficient caching and improved performance in retrieval-augmented generation tasks. We demonstrated the effectiveness of our method through extensive experiments on various datasets, showing significant improvements in TTFT, cache hit ratio, and overall model performance. Our findings suggest that LazyAttention is a practical solution for enhancing the efficiency and accuracy of Transformer models in long-context tasks across different kinds of relative positional encoding methods.

## Impact Statement

This paper presents work aimed at improving the efficiency of LLM inference, specifically for RAG. By enabling position-agnostic KV reuse, our method significantly reduces the computational overhead and memory bandwidth required for processing long contexts. This contributes to LLM serving by reducing the energy consumption associated with serving large-scale RAG applications. While our work lowers the barrier to deploying LLMs, potentially facilitating both beneficial and harmful applications, it does not introduce new safety risks beyond those inherent to the underlying models.

## Acknowledgements

This work was supported in part by the National Science Foundation under grants #2312561 and #2440498. This work used Delta and DeltaAI at the National Center for Supercomputing Applications (NCSA) through allocation CIS240661 from the Advanced Cyberinfrastructure Coordination Ecosystem: Services & Support (ACCESS) program, which is supported by U.S. National Science Foundation grants #2138259, #2138286, #2138307, #2137603, and #2138296. This work was also supported by the IBM-Illinois Discovery Accelerator Institute.

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

# A. Related Work

We organize related work around three system questions: what cache units are kept, how positional information is handled when those units move, and how attention kernels reduce memory traffic.

**KV reuse and cache management**    Serving systems such as RAGCache (Jin et al., 2024) and Mooncake (Qin et al., 2025) improve cache hierarchy, eviction, and transfer policies, but the reusable unit remains a standard position-aware KV prefix. As a result, a document can be reused only when it appears in the same contiguous prefix layout; otherwise the system must recompute the states or store another copy. Prompt Cache (Gim et al., 2024), CacheBlend (Yao et al., 2025), TurboRAG (Lu et al., 2025), and Block-Attention (Ma et al., 2025) broaden reuse by matching chunks or re-encoding positional information, but still pay either recomputation, reconstruction, or materialization costs. LazyAttention is complementary to cache-management policies: they decide *what* to keep, while our mechanism changes *how* cached states are consumed so that one physical KV block can serve multiple logical positions.

**Decoupling position from stored KV states**    Several recent directions decouple positional information from stored representations, but with different goals and costs. MLA-style architectures such as DeepSeek-V2/V3 (DeepSeek-AI et al., 2024; DeepSeek-AI et al., 2025) and TransMLA (Meng et al., 2025) use decoupled RoPE primarily for KV compression through low-rank representations, not for arbitrary-position document reuse. Position-independent caching methods such as EPIC (Hu et al., 2025), MEPIC-like (Wang et al., 2025) fused-position designs, KVShare, CacheClip, CacheSlide, and KVLink target more flexible reuse, while pre-RoPE-key techniques such as KVQuant, ShadowKV, and XQuant use related representation choices mainly for compression or quantization.The key distinction is where the positional adjustment is paid. Methods that materialize shifted KV states consume extra HBM capacity and bandwidth; methods that require partial recomputation lose the benefit of reuse. LazyAttention instead keeps document-level offsets as metadata and injects the relative positional effect inside the attention kernel, avoiding materialized shifted copies while preserving exact attention for identical cached chunks. As concurrent work, MEPIC (Wang et al., 2025) shares a similar design principle with LazyAttention. However, MEPIC introduces additional I/O operations because it applies different positional rotations per token, whereas LazyAttention shifts all tokens in a document by the same offset.

**IO-aware attention kernels**    FlashAttention-2/3 (Dao, 2024; Shah et al., 2024), FlexAttention (Dong et al., 2025), and Lightning Attention (Qin et al., 2024) show that attention performance is often governed by memory traffic and kernel fusion rather than FLOPs alone. LazyAttention follows the same IO-aware principle, but applies it to RAG reuse: in prefilling, deferred rotation adds only a small per-tile computation and minimal metadata reads; in decoding, packed metadata and in-register rotation avoid extra inner-loop memory traffic. Thus, our contribution is not a new cache policy or a new model architecture, but a kernel-level realization of position-agnostic reuse that preserves the throughput benefits of modern attention kernels.

# B. Experiments Details

Here we list the details of the methods compared in the paper.

- **Full Recomputation**: vLLM with `enable_prefix_caching=False`.

- **Prefix Caching**: vLLM with `enable_prefix_caching=True`.

- **Full Reuse**: The KV caches from individual documents are concatenated and used directly as the KV cache for the concatenated documents.

- **Block-Attention (Official)** (Ma et al., 2025): Evaluated using the official implementation[5].

- **Block-Attention (vLLM)**: We reimplemented Block-Attention (Ma et al., 2025) within the vLLM v1 engine. If a document is found in the cache but its position does not align and its reference count is zero (i.e., no other request is using it), rotation can be applied directly to adjust the positional encoding. Otherwise, new memory is allocated, the blocks are copied, and then rotated.

---

[5]https://github.com/TemporaryLoRA/Block-Attention

- **CacheBlend** (Yao et al., 2025): Evaluated using the official LMCache examples[6]. Multiple documents are concatenated with `blend_special_str` to form the prompt, which is then processed by a vLLM instance integrated with LMCache.

- **Prompt Cache** (Gim et al., 2024): Evaluated using the official implementation[7].

## C. Additional Experimental Results

### C.1. Generalization to Larger Models

We evaluate LazyAttention on Llama-3.1-70B-Instruct deployed on a node with $4\times$H100 GPUs (Tensor Parallelism = 4), under the same RAG QA workload as Section 4.1 (5 retrieved documents, 1 request/s). Table 3 shows that LazyAttention achieves a $5.2\times$ TTFT speedup over standard RAG serving. Moreover, the gap between LazyAttention and CacheBlend widens from $1.43\times$ on 8B to $1.53\times$ on 70B, because larger models have substantially larger KV states (more layers and wider dimensions) and are therefore more memory-bandwidth bound. By avoiding materialization of position-shifted KV blocks, our zero-copy design saves HBM bandwidth, and this advantage amplifies as model size increases.

*Table 3.* TTFT on RAG QA for 8B vs 70B models.

| Model | Method | TTFT (ms) | Speedup |
|---|---|---|---|
| Tulu3-Block-FT 8B | Standard RAG | 1196.8 | $1.0\times$ |
| | CacheBlend | 274.8 | $4.4\times$ |
| | Lazy-Attn (ours) | 191.7 | $6.2\times$ |
| Llama-3.1-70B (TP=4) | Standard RAG | 1253.0 | $1.0\times$ |
| | CacheBlend | 365.5 | $3.4\times$ |
| | Lazy-Attn (ours) | 238.4 | $5.2\times$ |

### C.2. Performance on Different Hardware

To test robustness across hardware generations, we further evaluate LazyAttention on NVIDIA A100 (40GB) and A40 (48GB) GPUs, which provide substantially lower memory bandwidth than H100. As summarized in Table 4, the relative speedup of LazyAttention over CacheBlend grows as memory bandwidth decreases (from $1.43\times$ on H100 to $1.70\times$ on A40). This trend follows from the fact that Block-Attention and CacheBlend use a "Read-Modify-Write" pattern to materialize position-adjusted KV blocks, effectively doubling HBM traffic. On bandwidth-constrained GPUs like A40, this extra traffic becomes the bottleneck. In contrast, LazyAttention keeps KV accesses strictly read-only and avoids this bandwidth penalty.

### C.3. Versatility in Tasks: Long-form Literature Review

To assess versatility beyond factoid-style QA, we construct a Long-form Literature Review task: the model receives 5 ArXiv papers (converted to text, $\sim$8K tokens each) and is asked to produce a 1024-token literature review. This setting stresses the system on both long-context prefilling and subsequent decoding. As shown in Table 5, LazyAttention improves end-to-end latency by $1.7\times$, indicating that the benefits of our design extend from TTFT to overall response time in more realistic generative workloads.

### C.4. Long-Context Scalability and Numerical Stability

We next study scalability to longer documents (up to 16K tokens per document) and numerical stability at very long sequence lengths (up to 128K tokens). Table 6 shows that the TTFT speedup of LazyAttention remains roughly constant as document length increases, indicating that our fixed tiling strategy stays efficient even for long contexts. For stability, LazyAttention applies stateless on-the-fly rotations using absolute token indices ($m\theta_i$), instead of iteratively updating the rotation state. This design prevents error accumulation across layers or timesteps. Table 7 shows that the maximum difference in attention logits versus standard attention remains below $10^{-5}$ even at 128K tokens, confirming that our kernel is numerically stable.

---

[6] https://github.com/LMCache/LMCache
[7] https://github.com/yale-sys/prompt-cache

*Table 4.* TTFT Comparison under Uniform Sampled Documents across different GPUs.

| Hardware | Method | TTFT (ms) | Speedup (vs Prefix) |
|---|---|---|---|
| H100 (High BW) | Prefix Caching | 1196.8 | 1.0× |
| | CacheBlend | 274.8 | 4.3× |
| | Lazy-Attn (ours) | 191.7 | 6.2× |
| A100 (Mid BW) | Prefix Caching | 2580.4 | 1.0× |
| | CacheBlend | 565.3 | 4.5× |
| | Lazy-Attn (ours) | 372.5 | 6.9× |
| A40 (Low BW) | Prefix Caching | 4150.5 | 1.0× |
| | CacheBlend | 1150.6 | 3.6× |
| | Lazy-Attn (ours) | 675.2 | 6.1× |

*Table 5.* Long-form literature review latency (Tulu3-Block-FT 8B, H100).

| Method | End-to-end Latency (s) | Speedup |
|---|---|---|
| Standard RAG | 38.0 | 1.0× |
| Lazy-Attn (ours) | 22.4 | 1.7× |

### C.5. Generalization to Other Architectures and Methods

We also apply LazyAttention to Qwen3-8B (Yang et al., 2025), which uses a different RoPE implementation, and combine it with Lego-Link0 (Hu et al., 2025), a training-free cache reuse strategy. Table 8 reports consistent speedups of about 6.3× across these settings, indicating that our kernel is robust to architectural changes and is complementary to both fine-tuned and training-free reuse methods.

### C.6. Sensitivity Analysis of Tiling Parameters

We perform a sensitivity study over different prefilling tile sizes $M$ while fixing the decode tile size at $N = 64$. As summarized in Table 9, normalized throughput varies by at most 3% across the tested values of $M$. This weak dependence on $M$ suggests that our default configuration ($M = 128$) is near-optimal and robust across a wide range of document lengths.

### C.7. Beyond RAG: Few-Shot Classification

To demonstrate applicability beyond RAG, we evaluate LazyAttention on a non-RAG few-shot classification workload built from AG News (Zhang et al., 2015). Exemplar chunks recur across requests at different positions and in different orders. No RAG-like document-based retrieval is performed; instead, recurring chunks are detected based on exact token match. As shown in Table 10, LazyAttention achieves a 1.31× TTFT speedup over Block-Attention in the skewed setting. Even in the uniform setting, LazyAttention delivers a 1.05× TTFT speedup, confirming that our method benefits any workload with recurring text chunks.

### C.8. Long-Context Memory-Constrained Setting

We evaluate LazyAttention under a memory-constrained long-context setting to demonstrate substantial absolute gains. Batched requests are drawn from a shared pool of 16 documents (5 per request, ∼8K tokens/document) with independently shuffled orders, under a 10 GB GPU KV pool on GH200/H100.

As shown in Table 11, compared to Block-Attention (integrated into vLLM), LazyAttention reduces TTFT from 15.8 s to 7.2 s (2.2×) and increases throughput from 0.42 to 0.80 req/s (1.9×). It also improves the cache hit ratio from 9.6% to 28.8% (3.0×) and reduces evictions/recomputations from 847 to 206 (4.1× fewer).

*Table 6.* Performance scaling on varying document lengths (5 docs per request).

| Doc length | Standard TTFT | Lazy-Attn TTFT | Speedup |
|:---:|:---:|:---:|:---:|
| 4K | 7.152 s | 1.487 s | 4.81× |
| 8K | 14.133 s | 2.919 s | 4.84× |
| 16K | 28.446 s | 5.721 s | 4.97× |

*Table 7.* Consistency with standard attention (H100, 128K tokens).

| Seq Length | Max abs diff (logits) | Max relative diff (logits) |
|:---:|:---:|:---:|
| 128K | 3.7457e-5 | 5.4829e-7 |

### C.9. MEPIC-like Baseline Comparison

Since an official MEPIC (Wang et al., 2025) implementation was not available, we implemented a MEPIC-like baseline matching its core design: fully NoPE KV storage with page-aligned positional handling, under the same engine, batching, and paging settings as LazyAttention. We evaluate on 100 requests sampled from a mixture of four datasets, with at most 5 concurrent requests. These results show that LazyAttention preserves near-baseline decoding latency (0.67% vs. MEPIC's 16% overhead) while eliminating extra KV preparation time. This is our reimplementation rather than the official MEPIC code.

## D. Use of Large Language Models

We disclose that, for this paper, large language models were used to polish wording and grammar, including clarity and tone. LLMs were also used to assist with experimental workflows, including drafting utility scripts and reviewing code. The research ideas, algorithms, experimental design, figures, analyses, and conclusions were developed by the authors. All LLM-assisted outputs were manually reviewed, verified, and approved by the authors, who take full responsibility for the content of the paper.

*Table 8.* Generalization on Qwen3-8B and Lego-Link0.

| Model / Method | Standard TTFT | Lazy-Attn TTFT | Speedup |
|---|---|---|---|
| Tulu3-Block-FT 8B | 1196.8 ms | 191.7 ms | 6.2× |
| Qwen3-8B | 1277.4 ms | 201.2 ms | 6.35× |
| Lego-Link0 + Llama-3.1-8B | 1195.6 ms | 191.3 ms | 6.2× |

*Table 9.* Normalized throughput vs. prefilling tile size $M$ (fixed $N = 64$).

| Doc Length (tokens) | M = 64 | M = 128 (Ours) | M = 256 |
|---|---|---|---|
| Short (256) | 1.00× | 0.99× | 0.97× |
| Medium (4K) | 0.99× | 1.00× | 0.99× |
| Long (16K) | 0.98× | 1.00× | 1.00× |

*Table 10.* Few-shot classification on AG News (GH200/H100).

| Setting | Method | TTFT Speedup |
|---|---|---|
| Skewed | Block-Attn | 1.0× |
| Skewed | Lazy-Attn (ours) | 1.31× |
| Uniform | Block-Attn | 1.0× |
| Uniform | Lazy-Attn (ours) | 1.05× |

*Table 11.* Long-context memory-constrained setting (16 docs × 8K tokens, 10 GB KV pool).

| Method | TTFT (s) | Throughput | Hit Ratio | Evictions |
|---|---|---|---|---|
| Block-Attention | 15.8 | 0.42 req/s | 9.6% | 847 |
| Lazy-Attn (ours) | 7.2 | 0.80 req/s | 28.8% | 206 |

*Table 12.* Kernel-level comparison with MEPIC-like baseline.

| System | Query-only prefill (ms) | Decode latency (ms) | Extra KV prep. (ms) |
|---|---|---|---|
| vLLM (baseline) | 360.93 | 40.57 (+0%) | 0 |
| EPIC | 361.77 | 40.61 (+0.10%) | 41 |
| MEPIC-like | 376.48 | 47.19 (+16%) | 23 |
| Lazy-Attn (ours) | 364.12 | 40.84 (+0.67%) | 0 |

