# OpenReview forum: "LazyAttention: Efficient Retrieval-Augmented Generation with Deferred Positional Encoding"
_ICML.cc/2026/Conference — ICML 2026 regular_

### Official Review · Reviewer_mEgF · 2026-03-07

**Soundness:** 4
**Presentation:** 4
**Significance:** 3
**Originality:** 3
**Overall Recommendation:** 4
**Confidence:** 3

**Summary:**

The paper introduces LazyAttention, an optimized attention mechanism designed to improve the efficiency of KV cache reuse in long-context scenarios like Retrieval-Augmented Generation (RAG) and In-Context Learning (ICL). Standard KV caches are position-aware, meaning that they must be recomputed or duplicated if a document appears at a different position in a new prompt. The authors propose Deferred Positional Encoding, which stores position-agnostic KV caches and only applies positional information lazily within the attention kernel during the computation phase. By kernelizing this process in Triton, LazyAttention avoids the memory overhead of duplicating caches for different document orderings while maintaining mathematical equivalence to standard attention. The system achieves a 1.37$\times$ reduction in TTFT and a 1.40$\times$ increase in throughput compared to state-of-the-art Block Attention in skewed document distributions.

**Compliance With Llm Reviewing Policy:**

Affirmed.

**Final Justification:**

**My final recommendation is 4: Weak Accept**.

I find the paper technically sound, practically significant, and clearly presented. Its main strength is tackling an important systems problem in LLM serving: reducing the cache-space barrier caused by document-order variation. The proposed LazyAttention framework is practically valuable, and the kernelized implementation makes the contribution more convincing than a purely conceptual design. I also appreciate the generally clear writing.

At the same time, I view the originality as somewhat more incremental than fundamental. The key idea of decoupling RoPE from KV caching is related to concurrent or recent directions, so the novelty lies more in the systems realization and optimization than in a new theoretical insight.
The authors’ rebuttal was helpful and addressed my main concerns sufficiently. In particular, the additional clarification and evidence strengthened my confidence that the method is well-engineered and practically useful, even if some limitations in novelty and evaluation scope remain. As a result, the rebuttal improved my assessment and made me more positive about acceptance.

**Overall, I believe the paper offers meaningful practical value to the community, and its strengths outweigh its remaining weaknesses. I also consider other reviewers' feedback and the rebuttal comments from the authors. Hence, I will keep my positive score.**

**Key Questions For Authors:**

**I would be willing to raise my score if the authors can (partly) address the following questions:**

- The authors mention the feasibility of caching a medium-sized database in a combination of HBM and host memory. Does the 0.2% overhead hold when documents are fetched from host memory, or does the latency of host-to-device transfers negate the benefits of the lazy kernel?

- The authors state that LazyAttention is mathematically identical to existing duplication methods. Can you clarify if there are any numerical stability issues or accumulation errors introduced when performing RoPE rotations inside the fused kernel at high sequence lengths (e.g., 128K tokens)?

- How does the complexity of the Triton kernel scale as the number of logical requests sharing a single physical KV copy increases within a single batch? Is there a limit to how many concurrent positions one physical document can support before the kernel becomes compute-bound?

**Meanwhile, I might have misunderstood some aspects of the paper, so I would welcome the authors’ clarification where appropriate.**

**Limitations:**

Yes

**Strengths And Weaknesses:**

**Strengths:**

1. The paper tackles a real-world problem in LLM serving. As context windows grow, the $O(N!)$ space complexity required for 100% cache hits with varying document orders becomes a hard wall. LazyAttention’s reduction of this to $O(N)$ is a major practical win for RAG systems.

2. The decision to kernelize the RoPE-decoupling rather than just proposing it as a high-level idea is commendable. The authors identify that the bottleneck of previous re-encoding attempts was memory materialization; by moving this logic on-chip during the attention loop, they bypass the I/O bottleneck.

3. The runtime overhead is remarkably low. This suggests that the custom Triton kernels are highly optimized and that the added logic for deferred encoding does not significantly interfere with the compute-bound nature of prefilling or the bandwidth-bound nature of decoding.

4. The paper is generally well written. The motivation is clear, the method section is easy to follow, and the figures help explain the design choices.

**Weaknesses:**

1. While the kernelization is a strong contribution, the idea of decoupling RoPE from the KV cache has been explored in concurrent or very recent work (e.g., TurboRAG, Block-Attention). The originality here is largely an engineering and implementation-level optimization. The theoretical novelty is somewhat limited.

2. The results are strongest for skewed document distributions. It would be beneficial to see how the system performs in a worst-case scenario where document reuse is minimal or when the KV cache is stored in host memory (off-chip) vs. HBM, as the paper mentions host memory as a potential storage layer.

3. The paper focuses heavily on RoPE. While RoPE is the current standard for models like Llama and Qwen, the paper could be more robust by discussing how (or if) this lazy approach applies to newer architectural shifts, such as ALiBi or linear attention variants, which might not be as easily deferred in a kernel (as I know).

---

> ### Author Rebuttal · Authors · 2026-03-31
>
> Thank you for the positive review. We clarify novelty and add evidence on low-reuse, host-memory-backed, long-context, and multi-sharing settings below.
>
> **W1 (On novelty).** Prior work (e.g., BlockAttention, TurboRAG) also explores RoPE decoupling at a high level, but the key difference is how it is realized and what system consequence it enables. Existing methods still rely on RoPE re-encoding/materialization in HBM, whereas LazyAttention eliminates RoPE in HBM and restores the exact positional effect inside the attention kernel with very low overhead. This is not merely an engineering detail: it is what removes shifted-KV duplication and leads to the cache-hit advantage in both theory and experiments. Empirically, the official Block-Attention implementation is 1.7× slower in TTFT than standard vLLM, and even after integrating BlockAttention into the same vLLM stack, LazyAttention still achieves 1.37× lower TTFT and 1.40× higher throughput.
>
> **W2 (On low-reuse / host-memory-backed settings).** We agree these settings are important.
> - In the zero-reuse case, we simply dispatch the regular attention kernel, so there is no additional overhead. Even when we force deferred positional encoding inside the GPU kernel, the overhead remains negligible.
> - The low-reuse regime is partially covered by the uniform-traffic setting in Table 1, where reuse is much scarcer. LazyAttention remains competitive with Block-Attention and still outperforms standard vLLM serving.
> - In the host-memory-backed setting, LazyAttention still avoids shifted-KV materialization after transfer. With 25% / 50% / 75% offloaded KV, it still reduces TTFT by 1.33× / 1.24× / 1.17×.
>
> **W3 (On broader applicability).** LazyAttention is not tied to standard RoPE: it applies whenever positional effects are injected into attention score computation rather than materialized into stored KVs. This includes RoPE-family variants and, in principle, score-space relative-position methods such as ALiBi or learned relative biases. Linear attention is outside our current scope, and we will clarify this boundary in the revision. A more through discussion can be found answer for Reviewer  mdMg W2&Q1.
>
> **Q1 (On host-memory overhead).** The reported 0.2% is the on-GPU, kernel-local overhead when KV is already resident on GPU; it excludes host-to-device transfer. With host-memory fetch, transfer dominates end-to-end TTFT, so the fractional overhead attributable to LazyAttention becomes even smaller, e.g., 0.17% under 50% offload.
>
> **Q2 (On long-context stability).** In short, ***no issues*** by design, which we also have tested empirically.
>
> Our LazyAttention applies stateless on-the-fly rotations using *absolute token indices* rather than iteratively updating a rotation state. We also verified this empirically up to 128K tokens in Appendix D.4, where the maximum absolute difference in attention logits relative to standard attention remains below 1e-5.
>
> **Q3 (On shared-KV scaling).** Let us address this question in two parts: ***(Part-I)*** a usual scenario where novel documents still appear occupying HBM, and ***(Part-II)*** a hypothetical scenario where only a single document is used across all requests.
>
> ***Part-I: Novel documents still appear:***  Sharing one physical KV copy across multiple logical requests does not change the asymptotic attention compute: each logical request still performs its own attention. The gain is eliminating the extra copy/re-encode/materialization cost incurred by prior position-dependent reuse methods. We directly vary the number of logical requests sharing one KV copy:
>
> | Number of sharers | Block-Attention TTFT (ms) | LazyAttention TTFT (ms) | Speedup | LazyAttention overhead |
> |-:|-:|-:|-:|-:|
> | 2 | 287 | 219 | 1.31× | 0.19% |
> | 4 | 533 | 401 | 1.33× | 0.20% |
> | 8 | 1012 | 761 | 1.33× | 0.21% |
>
> These results show that increasing the number of sharers does not introduce additional copy/materialization cost beyond normal batched attention.
>
> ***Part-II: Only one document appears in multiple requests:***
> We also test the extreme case where all requests share the same 16K-token document KV cache. Here, both the baseline (official Triton vLLM) and LazyAttention reuse the same precomputed document KV, so neither method needs to prefill the document itself; prefilling only handles a minimal query, followed by short decoding.
>
> | Logical requests | Baseline TTFT (ms) | Baseline TPOT (ms) | Baseline tok/s | Lazy TTFT (ms) | Lazy TPOT (ms) | Lazy tok/s |
> |---:|---:|---:|---:|---:|-:|-:|
> | 128 | 599.18 | 201.01 | 476.98 | 623.33 | 197.36 | 472.16 |
> | 256 | 981.23 | 368.94 | 525.74 | 847.22 | 366.41 | 515.16 |
> | 512 | 1640.52 | 707.84 | 554.24 | 1237.14 | 684.16 | 549.32 |
> | 1000 | 2775.34 | 1316.77 | 576.29 | 2147.91 | 1275.06 | 565.64 |
>
> In this extreme setting, LazyAttention is only slightly worse at 128 requests (623.33 vs. 599.18 ms TTFT), but becomes clearly better from 256 requests onward, while TPOT remains close or slightly better.

---

> > ### Author Rebuttal · Reviewer_mEgF · 2026-03-31
> >
> > Thank you for providing thorough answers to my concerns and questions.
> >
> > The additional clarification on novelty is helpful.
> >
> > I also appreciate the new results on low-reuse, host-memory-backed, long-context, and multi-sharing settings. These directly address my main concerns about worst-case behavior, host-memory overhead, numerical stability at long context lengths, and scaling when many logical requests share the same KV.
> >
> > Overall, the rebuttal meaningfully strengthens the paper and resolves several of my main concerns. I will keep my positive rating.

---

> > > ### Author Response · Authors · 2026-04-02
> > >
> > > Thank you for your positive acknowledgement! We truly appreciate your careful reading and are glad that the earlier evidence helped address your concerns.
> > >
> > > To make the overhead sources for ***Q1*** more transparent, we added a stage-wise breakdown on GH200/H100, covering host-to-GPU transfer, prefilling, and decoding. The setup uses one request with four 4096-token documents: under 0% offload, all four documents reside in HBM, while under 50% offload, two hot documents remain in HBM and two cold documents are placed in host memory.
> > >
> > > | Setting | cold doc 1 | cold doc 2 | hot doc 1 | hot doc 2 | decode token 1 (prefill) | decode token 2-9 (per token) | Overhead |
> > > |---|---:|---:|---:|---:|---:|---:|---:|
> > > | w/o Deferred Rotation, 0% offload | 0 | 0 | 0 | 0 | 353.92ms | 113.21ms | / |
> > > | with Deferred Rotation, 0% offload| 0 | 0 | 0 | 0 | 355.40ms | 113.39ms | 0.23% |
> > > ||
> > > | w/o Deferred Rotation, 50% offload | 5.51ms | 5.17ms | 0 | 0 | 349.23ms | 112.69ms | / |
> > > | with Deferred Rotation, 50% offload | 5.11ms | 5.26ms | 0 | 0 | 351.17ms  | 112.78ms | 0.18% |
> > >
> > > If any point still remains unclear, we would be very happy to clarify further. If this additional evidence strengthens your assessment of the paper, we would be grateful if you would kindly reflect that in your final rating.

---

### Official Review · Reviewer_H2EF · 2026-03-12

**Soundness:** 3
**Presentation:** 2
**Significance:** 2
**Originality:** 3
**Overall Recommendation:** 4
**Confidence:** 3

**Summary:**

This paper proposes Lazy-Attention to enable position-agnostic KV cache reuse for retrieval augmented generation. It stores KV without positional encoding and injects rotary offsets inside a fused attention kernel. It aims to avoid materializing position-shifted KV blocks while keeping attention results equivalent to re-encoding based methods. It reports large TTFT reductions on RAG workloads with small kernel overhead in practice.

**Compliance With Llm Reviewing Policy:**

Affirmed.

**Final Justification:**

Since my major concerns are resolved, I have updated the score.

**Key Questions For Authors:**

See weaknesses

**Strengths And Weaknesses:**

### Strengths
S1 The paper targets a practical bottleneck in RAG serving. It focuses on position coupling in RoPE based KV caching.

S2 The method is simple at the abstraction level. It moves positional adjustment into the attention kernel and keeps KV storage position free.

S3 The implementation details are concrete. The paper reports low overhead and provides an optimized kernel design for both prefill and decode.

S4 The evaluation shows strong TTFT gains across models and GPUs. It also reports robustness trends when bandwidth is lower.

### Weaknesses

W1 Novelty is not well positioned against closely related position-independent caching (PIC) systems. E.g. [MEPIC](https://arxiv.org/abs/2512.16822) also stores NoPE KV and applies rotary offsets inside a fused RoPE attention operator for position independent reuse. The paper does not cite or compare with this relationship.

W2 The baseline set misses a key PIC reference. EPIC is a well known position independent caching approach and should be compared directly.

W3 Related work on “pre-RoPE” KV handling is incomplete. [KVQuant](https://arxiv.org/pdf/2401.18079) first introduced pre-RoPE key quantization as a named technique. Later works such as [ShadowKV](https://arxiv.org/pdf/2410.21465) and [XQuant](https://arxiv.org/abs/2508.10395) also build on pre-RoPE keys in their designs. These works do not solve the same system problem, but they are important for terminology and historical positioning of the idea.

W4 The paper should discuss document level KV reuse systems that adjust positional embeddings at inference time. [KVLink](https://arxiv.org/abs/2502.16002) is relevant and currently not covered.

---

> ### Author Rebuttal · Authors · 2026-03-31
>
> Thank you for the careful review. Below, we clarify our relationship to prior PIC systems, pre-RoPE/NoPE work, and document-level reuse systems and add a new direct baseline comparison.
>
> **W1 (On novelty relative to PIC systems).** We thank the reviewer for highlighting MEPIC, which is concurrent with our work. Since an official MEPIC implementation was not available to us during rebuttal, we implemented and evaluated a ***MEPIC-like*** baseline that matches its core design choices as closely as possible: fully NoPE KV storage with page-aligned positional handling, under the same engine / batching / paging setting as LazyAttention.
>
> We evaluate on 100 requests sampled from a mixture of the four datasets used in the main paper, with at most 5 concurrent requests at any time:
>
> | System | KV form seen by attention kernel | Query-only prefilling latency | Decode latency | Extra KV preparation |
> | - | - | - | - | - |
> | vLLM (baseline) | Pre-materialized RoPE KV | 360.93 ms | 40.57 ms (0%)| 0 ms |
> | EPIC | Pre-materialized RoPE KV | 361.77 ms | 40.61 ms (+0.099%) | 41 ms |
> | MEPIC-like | Lazy KV handling (stores fully NoPE KV) | 376.48 ms | 47.19 ms (+16%) | 23 ms |
> | LazyAttention (ours) | Lazy KV handling (stores KV in local positions) | 364.12 ms | ***40.84 ms (+0.67%)*** | 0 ms |
>
> ***Note.*** For all methods, including the vLLM baseline, the full document KV is already cached. Thus, “Query-only prefilling latency” measures only the prefilling cost of the user query on top of cached document KV. “Extra KV preparation” denotes preprocessing before the attention kernel (e.g., block reconstruction, page alignment, or positional materialization), reported separately from decode latency.
>
> These results suggest that, relative to the MEPIC-like design, LazyAttention preserves near-baseline decoding latency while eliminating extra KV preparation. We will add this comparison in the revision and clearly state that this is our MEPIC-like reimplementation rather than official MEPIC code.
>
> ---
>
> **W2 (On PIC references and PIC-style positioning).** We agree that PIC-style references and positioning should be more complete, including EPIC, MEPIC, KVShare, and CacheClip.
>
> Our intended distinction is not simply “PIC vs. non-PIC,” but when and how positional information is materialized for KV handling. We organize this design space along two dimensions:
> (1) serving phase: prefill vs. decode;
> (2) positional overhead: recompute/materialize vs. avoid recompute/materialization.
>
> Specifically, EPIC requires runtime KV adjustments, while the concurrent MEPIC adopts a "lazy" fused-RoPE approach on NoPE representations. LazyAttention extends this "lazy" philosophy via a ***local-position*** design: by maintaining document-level offsets and injecting relative differences only within the decode kernel, we bypass fine-grained runtime handling. This ensures near-baseline memory traffic and negligible decode overhead. We will formalize this taxonomy in the revision.
>
> ---
>
> **W3 (On pre-RoPE / NoPE related work).** We agree that this discussion should be broadened. A taxonomy is:
>
> | Family | Subcategory | Main idea | Representative works / models | Relation to ours |
> | - | - | - | - | - |
> | RoPE-based models | Upfront / pre-materialized RoPE KV | RoPE-transformed KV is fully materialized before cache reuse or manipulation | CacheBlend, EPIC, KVShare, CacheSlide, CacheClip, BlockAttention, TurboRAG, KVLink | Different: these methods assume RoPE is already materialized |
> | RoPE-based models | Lazy RoPE handling | RoPE materialization is deferred until needed by the serving path or attention kernel | MEPIC; ***ours (LazyAttention)*** | Closest category to ours |
> | Pure NoPE models | — | Remove explicit positional encoding altogether | TinyLlama-NoPE, NoPE-GPT-Small-Base | Different: we remain RoPE-based |
> | RoPE/NoPE hybrid models | — | Mix RoPE and NoPE layers / attention patterns within one model | *Rope to Nope and Back Again*; Llama 4; SmolLM3 | Related at the positional-formulation level, but different from our lazy KV handling focus |
>
> Many prior systems assume upfront-materialized RoPE KV, whereas MEPIC and our method are better understood as lazy RoPE handling approaches. We also agree that the terminology/history around “pre-RoPE” should be clarified. In particular, KVQuant introduced pre-RoPE key quantization, and later works such as ShadowKV and XQuant also build on pre-RoPE keys. These works target compression / quantization rather than arbitrary-position KV reuse, but they are relevant for historical positioning and will be cited explicitly in the revision.
>
> ---
>
> **W4 (On KVLink and document-level reuse systems).** We agree that KVLink should be discussed. In our taxonomy, it is a document-level reuse method on pre-materialized RoPE KV, rather than a lazy RoPE handling approach like MEPIC or LazyAttention.

---

> > ### Author Rebuttal · Reviewer_H2EF · 2026-03-31
> >
> > Thanks for the response. Generally, the authors did a good job providing additional experiments to ease my concern.
> > I also learnt that the MEPIC is concurrent (Dec 2025), which relieves my primal concern on novelty.
> > Hence, I would like to raise my score.

---

> > > ### Author Response · Authors · 2026-04-02
> > >
> > > Thank you for your updated rating and positive acknowledgement!
> > >
> > > To make our positioning relative to prior PIC-style systems more explicit, we also summarize the design space below along two axes: whether KV can be shared across positions using a single physical copy, and whether reuse still requires recomputing KV of some tokens during prefill-stage.
> > >
> > > |                  | **Duplication**                                          | **Zero-copy**            |
> > > | ---------------- | -------------------------------------------------------- | ------------------------ |
> > > | **No recompute** | [Block-Attention](https://openreview.net/forum?id=7zNYY1E2fq), [TurboRAG](https://aclanthology.org/2025.emnlp-main.334/)                                | ***ours (LazyAttention)*** |
> > > | **Recompute**    | [CacheBlend](https://arxiv.org/abs/2405.16444), [EPIC](https://openreview.net/forum?id=qjd3ZUiHRT&noteId=nEjRCvdG9t), [KVLink](https://openreview.net/forum?id=oDcAGSXZZP&referrer=%5Bthe%20profile%20of%20Shiyu%20Chang%5D(%2Fprofile%3Fid%3D~Shiyu_Chang1)), [KVShare](https://arxiv.org/abs/2503.16525), [CacheSlide](https://www.usenix.org/conference/fast26/presentation/liu-yang), [CacheClip](https://arxiv.org/abs/2510.10129v1) | [MEPIC](https://arxiv.org/abs/2512.16822)                   |
> > >
> > >
> > > If any point still remains unclear, we would be very happy to clarify further. If this clarification is helpful, we would appreciate it if it can be taken into account in your final assessment.

---

### Official Review · Reviewer_mdMg · 2026-03-12

**Soundness:** 4
**Presentation:** 3
**Significance:** 3
**Originality:** 2
**Overall Recommendation:** 4
**Confidence:** 3

**Summary:**

Positional-adjusted KV caches limit reuse efficiency in retrieval-augmented generation (RAG), as existing approaches, such as Block Attention, still require KV duplication. This paper introduces LazyAttention, an attention mechanism with optimized Triton kernels for zero-copy, position-agnostic reuse of KV caches. The core idea is to defer rotary positional encoding (RoPE) and apply the necessary relative rotations directly inside the prefilling and decoding attention kernels, thereby avoiding the materialization of position-adjusted KV copies in HBM. The authors present a tiling-based analysis showing only constant-factor overhead and implement specialized kernels for both prefilling (compute-bound) and decoding (bandwidth-bound). Empirically, the method improves cache hit ratio by 7.2× over naive prefix caching with only ~0.2% overhead, while reducing time-to-first-token (TTFT) by 1.37× and increasing inference throughput by 1.40× compared to the Block Attention baseline.

**Compliance With Llm Reviewing Policy:**

Affirmed.

**Final Justification:**

This paper addresses the position dependency of the KV cache reuse via deferred RoPE application.

The rebuttal adequately addressed my concerns. Notably, the Zipf-based framing is more realistic than the original factorial argument. Preliminary ALiBi results on Falcon-7B add concrete evidence for generalizability beyond RoPE. The non-contiguous reuse discussion was thorough.

The contribution of this paper outweighs its weaknesses, but given the limited originality, I maintain my score of 4 (weak accept).

**Key Questions For Authors:**

1. How does LazyAttention generalize to other RoPE variants and deployment settings, such as interleaved RoPE (e.g., Llama), NTK-scaled RoPE, and GQA/MQA attention layouts? Are any kernel changes or special handling required?

2. How does LazyAttention handle non-contiguous or partially reused spans within the same document, such as two disjoint blocks with different positional offsets? In this case, does the $r = \frac{1}{B}$ assumption still apply, and what is the worst-case overhead when offsets vary frequently?

**Limitations:**

Yes

**Strengths And Weaknesses:**

#### Strengths

1. The paper identifies position dependency as a fundamental limitation of existing KV reuse methods and proposes a principled deferred RoPE adjustment that enables true zero-copy, position-agnostic KV reuse.
2. It presents a strong systems co-design, implementing separate prefilling and decoding kernels with careful reasoning about where to apply rotations (K vs. Q) and how to fuse them efficiently into the attention inner loops.
3. The evaluation uses realistic serving scenarios, including both uniform and skewed document popularity distributions, and reports metrics such as TTFT vs. arrival rate, cache hit ratios under VRAM constraints, and kernel-level overhead. Results show consistent improvements across multiple baselines (Prompt Cache, CacheBlend, Block-Attention), while maintaining negligible decoding overhead (~0.13%).
5. The motivation for position-agnostic reuse is clearly illustrated with intuitive diagrams contrasting eager and deferred positional encoding and their impact on KV cache shareability.

---

#### Weaknesses

1. The factorial $O(N!)$ storage argument characterizing existing approaches is rhetorically strong but not realistic. In practice, systems would not pre-store all permutations, so framing this as the baseline may somewhat overstate the relative advantage.
2. The method is designed for RoPE-based models, and its applicability to other positional encoding schemes (e.g., ALiBi or learned relative biases) remains unclear. The paper briefly mentions potential extensions, but a more detailed discussion of how the approach would generalize to modified or interleaved rotary variants would strengthen the work.
3. The main results are reported on a single high-end GPU (H100 96GB). Although the appendix includes additional hardware and model results, the main paper would benefit from presenting broader multi-model and multi-hardware evidence more prominently.

---

> ### Author Rebuttal · Authors · 2026-03-31
>
> Thanks for your careful and encouraging review. We address the primary concerns regarding framing, scope, and broader applicability below.
>
> **W1 (On practical framing of the cache advantage).** The $O(N!)$ statement is for intuitive understanding.
>
> A practical analysis — with zipf document occurrences and fixed cache capacity — still shows a significant gap between ours and a baseline. That is, ***our technique offers ${ln(C)} / (ln(C)-ln(D))$ times higher cache hit ratio,*** where $C$ is the cache capacity and $D$ is the number of unique positions a document can appear at. See ***Example*** below for concrete numbers.
>
> ***Details:*** Let document occurrences (i.e., popularity) follow a zipf distribution: the best caching strategy is to cache the heavy hitters. LazyAttentio caches the top $C$ documents. The baseline caches the top $C/D$ documents. LazyAttention's cache hit ratio is the sum of the probabilities of a zipf distribution up to $C$ documents, which is (approximately) proportional to $ln(C)$. Likewise, the baseline's cache hit ratio is (approximately) proportional to $ln(C/D)$. Therefore, the ratio between those two cache hit ratios is ${ln(C)} / (ln(C)-ln(D))$.
>
> ***Example***: Each document may appear in any of 20 positions (D = 20). We can cache up to 100 documents (C = 100). Our LazyAttention achives 2.86 times higher cache hit ratio.
>
> We will use this more practical model in the introduction.
>
> ---
>
> **W2 & Q1 (On applicability beyond standard RoPE / GQA-MQA).** LazyAttention is not tied to standard RoPE. It applies whenever positional effects are injected into attention score computation, rather than materialized as position-adjusted KVs.
>
> For RoPE-family variants, including interleaved RoPE used by Llama, scaled/NTK variants, and YaRN, the score can be written as
> $s(q,k,m,n)=\langle R_\theta(m)q,\;R_\theta(n)k\rangle = q^\top R_\theta(n-m)k.$
> Thus, the kernel only needs token positions plus lightweight encoding metadata (e.g., scaling parameters); the overall kernel structure remains unchanged.
>
> GQA/MQA also require no special algorithmic change: they only change the mapping from query heads to shared KV heads, while LazyAttention changes how each $Q$-$K$ score is computed.
>
> The same principle also covers score-space relative-position methods such as ALiBi and, in principle, learned relative biases, where the score is modified by a relative-position-dependent bias term rather than by materializing position-adjusted KVs.
>
> We will clarify this applicability scope more explicitly in the revision.
>
> ---
>
> **Q2 (On non-contiguous or partial reuse).** This question covers two different parts.
>
> ***Part-I: Intra-document Text Changes.***
> First, if the prefix within a document changes, LazyAttention does not attempt cache reuse, because the resulting KV states are generally no longer equivalent. For example, in “The athlete grabbed the trophy and dropped *it*.” versus “The athlete grabbed the opportunity and dropped *it*.”, the KV states for *it* should not be reused, even though the local text overlap is high. Accordingly, LazyAttention enables reuse only when the cached unit itself (e.g., a chunk, passage, or document) is identical, rather than when only an interior span happens to match.
>
> ***Part-II: Inter-document Order Changes.***
> Second, inter-document order changes are exactly the setting LazyAttention is designed for. For example:
>
> - Req 1: [medical-doc-A, medical-doc-B, “What’s my disease? Answer only using the last document.”]
> - Req 2: [medical-doc-B, medical-doc-A, “Diagnose my condition. Only use the previous document.”]
>
> Here, each document KV is reused as an intact unit, while the attention kernel injects the appropriate positional offset at score-computation time. Thus, document A and B from Req 1 can be reused in Req 2 without materializing new position-shifted KV copies.
>
> More generally, if $B$ reused blocks are partitioned into $S$ contiguous reused segments with different offsets, the metadata ratio becomes $r=\frac{S}{B}$; the previously discussed $r=\frac{1}{B}$ is the special case $S=1$. In the worst case, offsets vary at every block, so $r=1$, in which case LazyAttention loses most of its reuse advantage but correctness remains unchanged.
>
> ---
>
> **W3 (On broader multi-model / multi-hardware evidence).** Our Appendix D includes results with various models and different hardware, which we will happily move to the main section of the paper. Specifically, our results include Llama-3.1-70B (4×H100), A100/A40 hardware generalization, numerical stability up to 128K tokens, and cross-model results on Qwen3-8B.

---

> > ### Author Rebuttal · Reviewer_mdMg · 2026-04-01
> >
> > I thank the authors for the response. The practical Zipf-based cache analysis is a much stronger framing than the original $N!$ argument. The clarification on RoPE variant and GQA/MQA compatibility is reassuring, though I would note the ALiBi extension remains "in principle". A brief discussion of any practical limitations would be welcome in the revision.
> >
> > I am satisfied with the rebuttal and maintain my score, as my original assessment already accounted for the method's strengths.

---

> > > ### Author Response · Authors · 2026-04-02
> > >
> > > Thank you for your positive acknowledgement! We also appreciate your helpful suggestion that the ALiBi discussion in our rebuttal still remained somewhat "in principle."
> > >
> > > To make this point more concrete, we implemented a preliminary ALiBi-compatible lazy attention path and evaluated it on [Falcon-7B](https://huggingface.co/tiiuae/falcon-7b), one of the few open-source models supported by vLLM that exposes ALiBi. Instead of injecting RoPE-style relative rotations inside attention, the modified kernel injects the corresponding ALiBi score-space bias while preserving the same position-agnostic KV reuse principle. In a preliminary ablation (2 retrieved documents per request, 768 tokens per document, 128 output tokens), we obtained the following results and would present this carefully in the revision:
> > >
> > > | Metric | Baseline (vLLM) | Lazy ALiBi | Change |
> > > |---|---:|---:|---:|
> > > | TTFT (ms) | 1996.92 | 308.05 | -84.6% |
> > > | TPOT (ms) | 17.22 | 17.22 | <0.06% |
> > >
> > > If any point still remains unclear, we would be very happy to clarify further. If this additional evidence strengthens your assessment of the paper, we would be grateful if you would kindly reflect that in your final rating.

---

### Official Review · Reviewer_qd18 · 2026-03-13

**Soundness:** 3
**Presentation:** 3
**Significance:** 3
**Originality:** 3
**Overall Recommendation:** 4
**Confidence:** 2

**Summary:**

This paper proposes LazyAttention, a new attention mechanism for RAG that enables zero-copy, position-agnostic KV cache reuse by deferring positional encoding into the attention kernel. The key idea is simple and practically meaningful: existing KV reuse methods are still position-aware, so they either only reuse prefixes or need to duplicate KV cache for re-encoding. LazyAttention resolves this by kernelizing deferred RoPE encoding, allowing one physical KV copy to be reused at different logical positions. This work addresses an important bottleneck in RAG serving, and the empirical results are strong, showing better TTFT, higher cache hit ratio, and comparable quality to prior reuse methods.

**Compliance With Llm Reviewing Policy:**

Affirmed.

**Final Justification:**

LazyAttention proposes a clean, practical idea — deferring positional encoding into the attention kernel for position-agnostic KV cache reuse. The systems effort is solid and the method addresses a real bottleneck in RAG serving.

The rebuttal fully resolved my concerns. The non-RAG few-shot classification experiment demonstrates applicability beyond document-reuse workloads, and the long-context experiments show substantial absolute gains (2.2× TTFT reduction) under memory-constrained settings, addressing my concern about moderate speedups over the strongest baseline.

I keep my score at 4 (weak accept). The paper is technically sound with clear practical value, though the contribution remains somewhat narrow in scope.

**Key Questions For Authors:**

1. How broadly does this method generalize beyond RAG-style document reuse workloads?

**Limitations:**

Yes.

**Strengths And Weaknesses:**

**Strengths**
1. Practical value.
The paper proposes a clean idea: make KV cache position-agnostic by deferring positional encoding into the fused attention kernel. This will not only tackle the existing serving issues of RAG, but also benefit some sparse attention methods which may need to re-encode the positional information of context tokens.
2. Strong empirical results.
The evaluation is convincing. LazyAttention improves TTFT and cache hit ratio over strong baselines such as Block-Attention, CacheBlend, and Prompt Cache, especially under skewed document reuse. It also shows negligible kernel overhead and comparable QA quality.
3. Good systems effort.
The method is implemented with separate Triton kernels for prefilling and decoding, and the paper does a good job analyzing why the overhead remains very small in both regimes.

**Weaknesses**
1. Evaluation is focused on RAG-style reuse scenarios.
This is the right target setting, but it would still be helpful to better understand how broadly the method applies beyond document-reuse-heavy workloads.
2. Speedup over the strongest baseline is moderate.
The gains over Block-Attention are meaningful, but not huge in absolute terms, around 1.37× TTFT reduction and 1.40× throughput improvement in the main setting.

---

> ### Author Rebuttal · Authors · 2026-03-31
>
> Thank you for the positive review. Below, we clarify the scope of LazyAttention and the significance of the measured gains. We hope these responses address your concerns.
>
> **W1 (On applicability beyond RAG).** Even beyond RAG, our method can offer cache re-use benefits (*with near-zero overhead*) if there is a chance that some chunks may appear again. To show this, we performed additional experiments:
> - Setup: A non-RAG few-shot classification workload is built from AG News [1]. Exemplar chunks recur across requests at different positions and in different orders. *No RAG-like document-based retrieval is performed; instead, recurring chunks are detected based on exact token match.* GH200/H100 is used.
> - Results: Our LazyAttention achieved **1.31×** TTFT speedup and **1.27×** throughput improvement over Block-Attention in the skewed setting. Even in the uniform setting, ours delivered a **1.05×** TTFT speedup.
>
> This shows that our LazyAttention applies beyond RAG and beyond document-reuse-heavy workloads: gains are largest when reuse is abundant, but the method remains beneficial even when reuse is less.
>
> ---
>
> **W2 (On the significance of the measured gains).** Our approach shines for long contexts (e.g., 8K tokens/doc), providing more than *8-second* reductions in ***absolute*** TTFT, as demonstrated by our new experiments below.
>
> - Setup: Batched requests are drawn from a shared pool of 16 documents (5 per request, ~8K tokens/document) but use independently shuffled orders under a 10 GB GPU KV pool on GH200/H100 (5120 blocks with 16-token blocks for our 8B setting).
> - Results: Compared to BlockAttention (even after our careful migration from torch to vLLM), our LazyAttention **reduces TTFT from 15.8 s to 7.2 s (2.2×) and increases throughput from 0.42 to 0.80 req/s (1.9×)**. It also **improves the cache hit ratio from 9.6% to 28.8% (3.0×) and reduces evictions/recomputations from 847 to 206 (4.1× fewer)**.
>
> This result shows that when HBM is limited, the gap to the strongest baseline becomes substantial not only in relative terms but also in absolute wall-clock latency.
> ### References
>
> [1] Xiang Zhang, Junbo Jake Zhao, Yann LeCun. Character-level Convolutional Networks for Text Classification. NIPS 2015: 649-657.

---

> > ### Author Rebuttal · Reviewer_qd18 · 2026-04-02
> >
> > The rebuttal addressed my concerns, and I am satisfied with the clarifications and additional experiments. I will keep my positive rating.

---

> > > ### Author Response · Authors · 2026-04-02
> > >
> > > Thank you for your positive acknowledgement! We truly appreciate the time and care you took to reconsider the paper. We will carefully take your comments into account in the revision.
> > >
> > > If any point still remains unclear, we would be very happy to clarify further.

---

### Decision · Program_Chairs · 2026-04-30

**Decision:**

Accept (regular)

**Comment:**

All reviewers expressed a positive overall assessment of the paper, particularly noting the strength of the system design and the empirical results. At the same time, they raised several questions regarding the broader applicability of the proposed approach, including its effectiveness beyond document-reuse-heavy workloads and its compatibility with alternative positional encoding schemes such as ALiBi. Based on the rebuttal and subsequent discussion, I believe these concerns were addressed satisfactorily. Accordingly, I am inclined to recommend acceptance.